# The Pancreatic Microbiome is Associated with Carcinogenesis and Worse Prognosis in Males and Smokers

**DOI:** 10.3390/cancers12092672

**Published:** 2020-09-18

**Authors:** Jaideep Chakladar, Selena Z. Kuo, Grant Castaneda, Wei Tse Li, Aditi Gnanasekar, Michael Andrew Yu, Eric Y. Chang, Xiao Qi Wang, Weg M. Ongkeko

**Affiliations:** 1Department of Surgery, Division of Otolaryngology-Head and Neck Surgery, UC San Diego School of Medicine, San Diego, CA 92093, USA; jchaklad@ucsd.edu (J.C.); gecastan@ucsd.edu (G.C.); wtl008@ucsd.edu (W.T.L.); agnanase@ucsd.edu (A.G.); 2Research Service, VA San Diego Healthcare System, San Diego, CA 92161, USA; 3Department of Medicine, Columbia University Medical Center, NY 10032, USA; szk9009@nyp.org; 4Department of Internal Medicine, Emory University School of Medicine, Atlanta, GA 30322, USA; michael.yu@emory.edu; 5Department of Radiology, University of California, San Diego, CA 92093, USA; e8chang@health.ucsd.edu; 6Radiology Service, VA San Diego Healthcare System, San Diego, CA 92161, USA; 7Department of Biomedical Sciences, College of Medicine, Florida State University, Tallahassee, FL 32306, USA; XiaoQi.Wang@med.fsu.edu

**Keywords:** pancreatic adenocarcinoma, microbiome, smoking, tobacco, gender, TCGA

## Abstract

**Simple Summary:**

The cancer microbiome has been suggested to be closely involved in the immune dysregulation that leads to carcinogenesis. Given that pancreatic adenocarcinoma (PAAD) is one of the most lethal cancers, it is important to identify features of the microbiome that may contribute to more deadly PAAD tumors. In this study, we analyzed PAAD patient RNA-sequencing data from The Cancer Genome Atlas (TCGA) to correlate abundance of intra-pancreatic microbes to dysregulation of immune and cancer-associated genes and pathways. We discovered that the presence of several bacteria species within PAAD tumors is linked to metastasis and immune suppression. Furthermore, we found that the increased prevalence and poorer prognosis of PAAD in males and smokers are linked to the presence of potentially cancer-promoting or immune-inhibiting microbes. Further study into the roles of these microbes in PAAD is imperative for understanding how a pro-tumor microenvironment may be treated to limit cancer progression.

**Abstract:**

An intra-pancreatic microbiota was recently discovered in several prominent studies. Since pancreatic adenocarcinoma (PAAD) is one of the most lethal cancers worldwide, and the intratumor microbiome was found to be a significant contributor to carcinogenesis in other cancers, this study aims to characterize the PAAD microbiome and elucidate how it may be associated with PAAD prognosis. We further explored the association between the intra-pancreatic microbiome and smoking and gender, which are both risk factors for PAAD. RNA-sequencing data from The Cancer Genome Atlas (TCGA) were used to infer microbial abundance, which was correlated to clinical variables and to cancer and immune-associated gene expression, to determine how microbes may contribute to cancer progression. We discovered that the presence of several bacteria species within PAAD tumors is linked to metastasis and immune suppression. This is the first large-scale study to report microbiome-immune correlations in human pancreatic cancer samples. Furthermore, we found that the increased prevalence and poorer prognosis of PAAD in males and smokers are linked to the presence of potentially cancer-promoting or immune-inhibiting microbes. Further study into the roles of these microbes in PAAD is imperative for understanding how a pro-tumor microenvironment may be treated to limit cancer progression.

## 1. Introduction

Pancreatic cancer is the 12th most common cancer and is highly lethal. The overall 5-year survival rate across all stages of pancreatic adenocarcinoma (PAAD) is 9%, although only 3% of patients with metastatic disease are alive after 5 years. PAAD is usually detected at an advanced stage, and most treatment regimens are ineffective, while current interventions to prevent, diagnose, and treat are not satisfactory, leading to the poor overall prognosis [1,2,3,4,5].

Pancreatic cancer is characterized by frequent mutations in KRAS, TP53, CDKN2A, and SMAD4, as well as the dysregulation of diverse signaling pathways, including TGF-β, Wnt/Notch, and hedgehog signaling [3,6,7]. Other unique features of pancreatic cancer are the presence of a microenvironment filled with immunosuppressive mediators and a dense stroma. Stromal cells promote tumor growth, invasion, metastasis, and chemoresistance. Acquired immune evasion, which is comprised of an immunosuppressive microenvironment, poor T cell infiltration, and a low mutational burden, is a dynamic entity connected to immune system control. Based on the special pathological traits of pancreatic cancer, novel strategies such as stoma-targeting therapy, immunotherapy, and neoantigen vaccines are emerging as treatments. However, these therapies face the challenge of overcoming the highly immunosuppressive tumor microenvironment in pancreatic cancer [7,8,9].

A novel component in shaping the immune system and affecting disease prognosis for pancreatic cancer is the microbiome. The microbiome is defined as the comprehensive genomic information encoded by the microbiota (bacteria, fungi, protozoa, and viruses) and its ecosystem, products, and host environment. The human microbiota offer protection from disease by maintaining nutrition and hormonal homeostasis, modulating inflammation, and detoxifying compounds [10,11]. An intra-tumoral microbiome signature was identified as predictive of the long-term survival of pancreatic cancer [12]. Recently, the microbiota has begun to be recognized as an important player in cancers. For example, the gut microbiota may act as a mediator of immune system activation, promoting cancer-associated inflammation and affecting tumor responses to therapies in multiple cancers [13]. For pancreatic cancer, studies demonstrated that the neoantigens of pancreatic cancer showed homology to microbial epitopes, suggesting that microbial factors may determine tumor behavior and patient outcomes [14]. On the other hand, the gut microbiome from a short-term survivor of pancreatic cancer promoted pancreatic tumor growth in a mouse model via increasing the infiltration of CD4+FOXP3+ and myeloid-derived suppressor cells. These studies suggest that both the intra-pancreatic microbiome and the gut microbiome may be implicated in the immunosuppression and development of PAAD [12,15].

In this study, we aimed to expand our current understanding of the relationship between PAAD and its intratumoral microbiome, using next-generation RNA-sequencing data from 187 PAAD patients. We identified microbes whose abundance was correlated with poor survival and metastasis. We investigated correlations between microbe abundance and the expression of cancer and immune-associated genes. Due to the importance of smoking and gender in PAAD progression, we identified microbes differentially abundant in males versus females and in smokers versus nonsmokers, and analyzed how these microbes correlated with cancer and immune pathways. We observed significant correlations between a high abundance of certain microbes and poor prognosis/immunosuppression. While previous studies have demonstrated connections between PAAD pathogenesis and microbial abundance in mice, our study performed the largest profiling of the intra-pancreatic microbiome in sample size, and we are the first to demonstrate immune and clinical variable associations with microbial abundance in human PAAD samples, to the best of our knowledge.

## 2. Results

### 2.1. Contamination Correction

To ensure that the microbiota extracted from The Cancer Genome Atlas (TCGA) data did not include environmental contaminants, rigorous methods were used to correct for potential contamination (Figure 1). Microbes well known to be common contaminants were removed first. The firmicutes, bacteroidetes, and fusobacterium phyla were removed entirely, as the bacteria belonging to these phyla are considered common contaminants. We inferred that microbes whose abundance does not increase as the total number of microbe reads sequenced increases are likely contaminants, as contaminants from the environment would affect all samples equally and would not be correlated with the total reads sequenced, while the intra-tissue microbe abundance will increase when the total reads sequenced increase. To identify such contaminants, we used the Spearman correlation of microbial abundance. We also examined individual sequencing plates to check if the variations in microbial abundance were based on unusually high abundance in a limited number of plates (Appendix A). In total, more than 200 taxa were identified as contaminants and, of these, 73 were identified by Pathoscope and were removed from the analysis. Bacteria removed based on a lack of abundance variation and the date of sequencing did not belong to a common phylum, family, or genera. A list of all the removed bacteria can be found in Appendix A.

### 2.2. Microbial Abundance Correlated with Poor Patient Prognosis

To determine that the microbes correlated with PAAD progression, we correlated the microbial abundance to metastasis and patient survival data. We observed that, for many microbes, a high microbial abundance was correlated to metastasis (Kruskal–Wallis, *p* < 0.05) (Figure 2). The majority of microbes with such correlations belonged to the *Proteobacteria* phylum. To identify other microbes correlated with poor PAAD patient prognosis, we used Kaplan–Meier survival curves to compare the microbial abundance to patient survival (Figure 2). Microbes were considered to be potential contributors to poor prognosis if high abundance was correlated to lower patient survival (Cox regression, *p* < 0.05). Interestingly, the microbes that were correlated to survival were not significantly correlated to metastasis. Due to previous research implicating the high abundance of pancreatic microbiome microbes with increased tumor progression, we focused subsequent analyses on microbes with a positive correlation between abundance and either metastasis or poor survival. Interestingly, few microbes were correlated with tumor histology. Only one microbe previously correlated with metastasis, *Acidovorax ebreus* TPSY, was correlated to a high tumor grade. Few other microbes, including those not previously correlated to metastasis, showed a connection to tumor grade. This is likely due to the small number of patients in the TCGA PAAD cohort that have high-grade tumors (Appendix A).

To identify the gene expression signatures associated with microbial upregulation in PAAD, we used Gene Set Enrichment Analysis (GSEA). Interestingly, a subset of pathway dysregulation in a select group of upregulated microbes was centered around methylation (Figure 3A). Specifically, *Aggregatibacter aphrophilus* NJ8700; the primary endosymbiont of *Sitophilus zeamais*; *Mycoplasma hyopneumoniae*; *Beutenbergia cavernae* DSM 12333; and *Agrobacterium radiobacter* K84 were correlated with the upregulation of methylation signatures, specifically H3K27Me3 methylation, and the downregulation of pathways associated with methylation, such as WNT/β-catenin signaling [16].

An analysis of cancer and immune pathways showed a large number of microbes correlated with immune suppression, the downregulation of tumor suppressive pathways, and the upregulation of oncogenic pathways (Figure 3B). *Citrobacter freundii* and uncultured *Pseudomonadales bacterium* Hf0500_12O04 abundance were correlated to the upregulation of proinflammatory immune pathways, such as the inflammasome pathways and genes upregulated in macrophages treated with LPS. *C. freundii* was also correlated to multiple immunosuppression and oncogenic pathways. The abundance of nine microbes was correlated to the downregulation of tumor suppressive pathways. *Toxypothrix sp.* PCC 7601; the primary endosymbiont of *S. zeamais*; *Acidovorax ebreus* TPSY; and *Shigella sonnei* Ss046 were correlated to the downregulation of signatures directly related to p53 or downstream of p53. *M. hyopneumoniae* was correlated with the upregulation of various oncogenic pathways, as well as with the suppression of immune pathways. The uncultured bacterium HF0500_10F10 abundance was correlated to the upregulation of growth factor and ERBB2-related signatures and the downregulation of a p38 signature. To further analyze the effects of high microbial abundance on immune cell populations, we used the Cibersortx tool to calculate the relative immune cell population sizes via related gene expression (Figure 3C). We found that A. ebreus was correlated with significant immune dysregulation. Specifically, high abundance was correlated with lower total M2 macrophage, activated memory T cell, and CD8 T cell population sizes, suggesting immune suppression.

### 2.3. Validation of Microbial Presence

To validate that the microbes found in TCGA samples are present in other pancreatic cancer microbiomes, we repeated the Pathoscope alignment method on data obtained from a pancreatic tissue sequencing data deposited at the Sequencing Read Archive (SRA) (GSE79670). Out of the 13 microbes correlated to metastasis in the TCGA dataset, Pathoscope identified eight microbes that were present in the validation dataset. Using the Qiime2 computational framework, we calculated the relative evenness and diversity of the validation cohort using various measures of alpha diversity. We found that the alpha diversity indices were similar between the TCGA and validation cohorts (Appendix A). However, there was a much larger range of diversity in the validation cohort than in the TCGA cohort. Specifically, the Simpson evenness index showed a much larger variance in evenness in the validation cohort. This may be due to differential methods for collecting and sequencing tissue between the TCGA and validation cohorts. The validation cohort was further limited in that it did not contain more detailed clinical data for the samples collected. Therefore, we could not use this cohort in our further studies of patient etiologies. Despite these limitations, the agreement between the TCGA dataset and the validation dataset indicates a degree of external validity to our analysis of TCGA samples.

### 2.4. Microbial Abundance Affected by Risk Factors

Previous studies have shown that there are significant impacts that certain behaviors and characteristics can have on the composition of the microbiome. Two main risk factors for PAAD are gender and smoking, with males and smokers being predisposed to pancreatic cancer. To understand how the prognoses of these two groups may be related to the microbiome, we stratified the PAAD patient data into cohorts based on these two etiologies (Figure 4A). We used a principle component analysis of beta diversity to visualize differences between our patient groups (Figure 4B,C).

### 2.5. The Male vs Female PAAD Patient Microbiota

To analyze the differences between male and female patients’ microbiota, we used the Kruskal–Wallis test to determine the differential abundance between the two groups (Figure 5A). *A. ebreus* and *Acinetobacter baumannii* SDF were significantly upregulated in male patients, while *Geobacillus kaustophilus* HTA426 and *Escherichia coli* 55989 were significantly upregulated in female patients (Kruskal–Wallis, *p* < 0.05). We also used GSEA to analyze the immune and cancer-associated pathway dysregulation associated with these gender-related microbes (Figure 5B,C). *A. ebreus* did not yield significant GSEA data due to the poor plot quality. *G. kaustophilus* abundance was correlated with the downregulation of tumor suppressive signatures and upregulation of immunological signatures in males, while *E. coli* and *A. baumannii* abundance showed the opposite trend and were correlated with the upregulation of oncogenic signatures and suppression of immune signatures. In female patients, *G. kaustophilus* abundance was correlated with the downregulation of oncogenic signatures and upregulation of immune signatures. This suggests that *G. kaustophilus* may have different effects depending on the patient gender. *E. coli* abundance seemingly has less of an effect on gene expression in females, as little pathway dysregulation was observed in comparison to males. *A. baumannii* abundance correlates to the downregulation of tumor-suppressive pathways, but does not show a significant correlation to immune suppression, which was seen in males. To further investigate how these microbes might affect gene expression, we used the REVEALER algorithm to correlate the microbial abundance to copy number alteration (CNA) and mutation rates (Figure 5D). *E. coli* abundance was correlated with CNA in both male and females, but the genomic loci of correlation did not have any overlap between the genders. No mutations were correlated with *E. coli* abundance. These results show that the pancreatic microbiome may have differing effects in male PAAD patients compared to female patients.

### 2.6. The Smoking vs Nonsmoking PAAD Microbiota

To analyze how the microbiome is altered by tobacco smoking, we used the Kruskal–Wallis test to compare the microbial abundance between smoker and nonsmoker PAAD patients (Figure 6A,B). *A. baumannii* and *M. hyopneumoniae* were the only two microbes significantly dysregulated between the two groups, both of which are upregulated in smokers. This could indicate that these microbes may be part of a pathway for tobacco to influence disease severity, since earlier in this study the abundance of both were positively correlated with indicators of carcinogenesis and worse patient outcomes. We used GSEA to further explore their correlations to cancer, immune, and methylations pathways in smokers (Figure 6C,D). Both microbes were correlated with the upregulation of oncogenic signatures, the downregulation of immune and tumor suppressive signatures, and significant methylation activity. The effects of these microbes on smoking patient gene expression aligns with our previous analyses of *A. baumannii* and *M. hyopneumoniae*. A REVEALER plot of *M. hyopneumoniae* indicates a significant correlation to deletions at the 9q13 locus, known for its potential tumor suppressive activity (Figure 6E). These data show that the effects of smoking may not alter the pancreatic microbiome in a widespread manner, but the few microbes that are dysregulated by smoking may contribute to worse PAAD outcomes. However, further research is required to determine the cause of microbe dysregulation.

## 3. Discussion

Pancreatic cancer has been demonstrated to interact with the intratumor microbiome in multiple recent studies [12,15], providing significant evidence of the existence of an intra-pancreatic microbiome that was previously unknown only a few years ago. Since the discovery of this microbiome, several studies in the past two years have attempted to characterize its origins and relationship to diseases. It has been suggested that the microbiota from the stomach, duodenum, biliary tract, and even esophagus could have gained access to the pancreas through the pancreatic duct [5,15]. However, studies have not agreed on various issues, including what the most likely route of pancreatic bacterial colonization is, whether the pancreas can be colonized under healthy conditions, and the profile of normal pancreatic flora [5,17]. Furthermore, many studies were performed in mice, which may not be relevant to human physiology. In this study, we profiled the intra-tumor pancreatic microbiome through large-scale sequencing data from TCGA, providing the largest comprehensive profiling of microbes in pancreatic tumor samples to date, to the best of our knowledge. Given the known influence of the intra-tumor microbiome on the development of cancers, the microbiota in the pancreas could be a key factor contributing to pancreatic cancer pathogenesis and progression.

The pancreatic microbiota have been associated with pancreatic ductal adenocarcinoma in only four studies. In Pushalkar et. al., mice experiments provided evidence for the existence of a pancreatic microbiome, and the presence of microbes promoted immunosuppression [12]. The elimination of microbes protected against invasive pancreatic cancer in mice. However, the results were validated using only 12 human pancreatic cancer samples. In Riquelme et. al., 68 pancreatic tumor samples were profiled, but no normal pancreas samples were examined [15]. In Thomas et. al., 16 pancreatic cancer samples and 7 normal pancreatic samples were examined [17]. However, they were not able to find microbiome composition differences between the cancer and normal samples. In Geller et. al., 65 pancreatic tumor samples were profiled [18]. While some of these studies profiled a large number of samples, none of the four studies investigated correlations between bacterial populations and gene expression dysregulations through mRNA sequencing, and none correlated bacterial abundance to clinical variables and immune phenotypes. These studies also yielded contradictory results. In this study, we extracted microbe reads from 187 pancreatic cancer samples and examined the relationship among microbe abundance, immunological changes, and gene expression signatures in an attempt to utilize this unprecedented large-scale analysis to address the myriad outstanding questions regarding the microbiome’s role in pancreatic cancer.

One of the most important questions surrounding the detection of microbes in pancreatic tissue is whether the detected microbes were contaminants from the environment during sample processing. Through three different analyses, we established that most of the microbes we discovered in our data were not contaminants, although we did identify some potential contaminants. While TCGA does not explicitly control for contamination in sample collection procedures, previous studies extracting microbe abundance from TCGA samples generally concluded that tissue-intrinsic and biologically relevant microbes could be extracted from TCGA data given the rigorous identification of contaminants [19,20].

Through correlations with clinical variables, we identified that a high microbial abundance of a number of bacterial species is correlated with poor prognosis, in terms of metastasis and survival. Microbe abundance is also generally associated with the high expression of methylation-related gene expression signatures, suggesting that microbe abundance may be associated with increased methylation. In accordance with Pushalkar et al., we found that a high microbial abundance was associated with immunosuppression, which includes the low infiltration of M2 macrophages and T-cells. High microbe abundance was also correlated with the activation of oncogenic pathways and downregulation of tumor suppressive pathways, possibly creating a pro-tumor microenvironment.

We were also able to find significant differences in microbe abundance between smokers and nonsmokers and between male and female patients. Smoking is a significant risk factor for pancreatic cancer that increases its risk by 75%, with a persistent increased risk even 10 years after quitting [21]. Gender is also a risk factor, with males being slightly more likely to develop pancreatic cancer than females, although it is unclear if this trend is due to the greater prevalence of smoking among men [21]. While we found provocative correlations between smoking/gender-associated microbes’ abundance and immune and cancer-associated pathways, more in vitro and in vivo experiments are required to elucidate the role of these microbes and their relationship to gender and smoking.

The most prominent microbes associated with clinical variables and immune pathways in pancreatic cancer are mostly from the phylum *Proteobacteria*. *Acidovorax ebreus*, a *Betaproteobacteria*, was correlated with poor prognosis and decreased immune infiltration. Members of the class *Gammaproteobacteria* are especially featured in our correlations. Their abundance correlates positively with increasing metastasis more than the abundance of any other class of microbes. *C. freundii* is associated with poor prognosis and the dysregulation of multiple immune and cancer-associated pathways. *Pseudomonadales* are associated with inflammasome activation and poor prognosis. *S. sonnei* abundance is associated with poor prognosis and the upregulation of cancer-associated pathways. Both *C. feundii* and *S. sonnei* are in the *Enterobacteriaceae* family and are known to be present in the human gut, sometimes as pathogens [22], suggesting that pancreatic bacteria are possibly translocated from the gut. Our results corroborate previous findings that show that *Proteobacteria* is the dominant bacterial species in pancreatic cancer, comprising almost half of all pancreatic bacteria [12]. It has been hypothesized that *Gammaproteobacteria* in the pancreas could promote chemotherapy resistance to the drug gemcitabine by metabolizing it [18], and our study is the first to demonstrate a concrete correlation between *Gammaproteobacteria* levels and pancreatic cancer prognosis.

## 4. Conclusions

Our study significantly advances the understanding of the pancreatic cancer microbiome composition and its relationship with clinical and immunologic variables. We also corroborated previously reported results on the intra-pancreatic microbiome. A total of 13 microbes were found to be correlated to advanced tumor progression. These microbes were correlated to the dysregulation of gene signatures related to oncogenic methylation, cancer progression, and immune system modulation. Of these 13 microbes, *A. baumannii* and *M. hyopneumoniae* were found to be correlated to smoking-mediated changes in the genome that cause PAAD. *A. ebreus*, *A. baumannii*, *G. kaustophilus,* and *E. coli* demonstrated differential abundance and activation of cancer and immune-associated pathways in male versus female PAAD patients. However, the effect of male versus female microbiota may also be attributed to other variables, such as hormone levels and differential biology based on gender. It would therefore be ideal to compare the microbiota of males and females with cancer to those of males and females without cancer. However, the few normal samples deposited at TCGA are tumor-adjacent normals. Therefore, the microbiota of cancer and normal samples may not be significantly different, because both types of samples come from the same patient pool. This presents a limitation of our study that must be resolved in the future. Through the validation of TCGA microbial abundance using a separate dataset of PAAD patients, we were able to prove that the microbes we found were not unique to only TCGA tumors. Deeper sequencing and in vitro and in vivo investigations of microbes relevant to pancreatic cancer may reveal potential diagnostic or therapeutic strategies through microbe DNA-based diagnostic panels or antibiotic regimens [18,20]. As one of the deadliest and most difficult to diagnose cancers, pancreatic cancer has defied many strategies to manage its highly lethal malignancy, making novel treatment or diagnostic options particularly attractive.

## 5. Materials and Methods

### 5.1. Data Acquisition from TCGA

Raw whole-transcriptome RNA-sequencing data for tumor tissue were downloaded from the TCGA legacy archive (https://portal.gdc.cancer.gov/legacy-archive/search/f) on 27 August 2019, for 187 PAAD samples. Level 3 normalized mRNA expression read counts for the above samples were downloaded from the Genomic Data Commons (GDC) portal (https://portal.gdc.cancer.gov/) on 5 August 2018. Clinical information for all the patients was downloaded from the Broad Institute’s GDAC Firehose (https://gdac.broadinstitute.org/). Genomic alteration information for each patient was obtained from the last analysis report (2016) of the Broad Institute TCGA Genome Data Analysis Center (http://gdac.broadinstitute.org/runs/analyses__latest/reports/). Between the time of data acquisition and the time of analysis by this study, none of these datasets have been altered or added too.

### 5.2. Extraction of Microbial Reads and Calculation of Microbial Abundance

Using the Pathoscope 2.0 program [23], the RNA-sequencing data were filtered for bacterial reads via direct alignment through a wrapper for Bowtie2. This framework utilizes a reference library to select for reads unique to organisms of interest. For this analysis, bacterial sequences deposited at the NCBI nucleotide database (https://www.ncbi.nlm.nih.gov/nucleotide/) were used as a reference library. Pathoscope generates two output measures quantifying the amount of bacterial species present in samples. One measure, best guess, quantifies the relative abundance of each species, expressed as a percentage. The other measure, best hit read numbers, signifies the absolute integer count of each species in the sequencing data.

### 5.3. Evaluation of Contamination Using Date of Sequencing

We applied a heuristic algorithm to extract the sequencing dates where this overexpression occurs, which allowed us to determine potential contaminants’ relationship with the sequencing date. We visualized the microbial abundance of cancer patients in the form of a heat map and removed any microbe where stretches of dates with high microbial abundance exist, which we identified as contamination. In other words, contaminants are marked by a non-uniform abundance across sequencing dates. For all the following analyses, we removed all the microbes that were identified as contaminants.

### 5.4. Evaluation of Contamination Based on Plates

The TCGA sequencing protocol includes the collection of tissue samples from multiple sites at a common sequencing center. Tissue samples are then sequenced on the same plates at such common centers. The information about which patient samples was sequenced on which plate is publicly available via the Broad Institute’s GDAC Firehose. We used this resource to group patients based on common sequencing plate IDs. The abundance values of microbes were associated with plates on which the samples were stored prior to sequencing using the Kruskal–Wallis test (*p* < 0.05) and the visual examination of abundance differences between different plates using a boxplot.

### 5.5. Evaluation of Contamination Using Microbial Abundance Counts

The abundance of individual microbes in each patient is plotted against the total microbe reads in the same patient to determine whether any microbe is likely a contaminant. The best hit results from Pathoscope are used for this analysis because absolute counts are required. In the resulting scatterplots, if a positive slope exists it is likely that the microbe was biologically relevant and physically present in the sample, since the counts per microbe increased with the number of microbes sequenced. If the scatterplot has a slope of close to zero and the counts of all the microbes are substantially above zero, it is likely that the microbe was a contaminant. This reasoning follows from the assumption that similar amounts of microbes will be present regardless of how many microbes are present in the tissue sample if the microbe is an environmental contaminant. The Spearman correlation test and the correlation coefficient (R^2^) were used to calculate the significance of a linear trendline and the slope of that trendline, respectively.

### 5.6. Evaluation of Contamination Using Previously Identified Contaminants

A list of phyla that have been previously determined to be present in DNA sequencing kits was obtained from a study by Glassing et al. [24]. A list of phyla that are common in the hospital setting was obtained from a study by Rampelotto et al. [25]. These two lists per used to identify bacteria as common contaminants.

### 5.7. Determination of Microbiome Diversity in Patient Samples

Using the Qiime2 framework, the best guess data output from Pathoscope were used to calculate the alpha diversity and beta diversity using the qiime diversity alpha and qimme diversity beta modules, respectively. Principle component analysis of the beta diversity results was performed via the qiime diversity pcoa module and visualized using the qiime emperor plot module, the latter of which uses the EMPeror tool.

### 5.8. Differential Microbial Abundance between Cancer Patients of Different Smoking Status and Gender

A differential abundance analysis was performed to compare the microbe abundance (percent abundance) in cancer tissues based on male vs. female and smoking vs. nonsmoking comparisons. Microbes that are present in less than 10 percent of the patients in a cancer cohort were excluded. The Kruskal–Walls analysis test was then applied to determine the differential abundance (*p* < 0.05).

### 5.9. Correlation of Microbial Abundance to Survival and Clinical Variables

Survival analyses were performed while using the Kaplan–Meier Model, with microbe expression being designated as a binary variable based on the presence or absence of microbes in tumor samples. Univariate Cox regression analysis was used to identify candidates that were significantly associated with patient survival (*p* < 0.05). Clinical variable analysis was performed using the Kruskal–Wallis test, as described above.

### 5.10. Correlation of Microbial Abundance to Immune Infiltration

The estimated relative immune cell infiltration levels for 22 cell types were computed using the software CibersortX [26]. The microbe abundance was then correlated with the immune cell infiltration levels for each microbe using the Kruskal–Wallis test (*p* < 0.05). Microbe abundance was modeled as a binary variable of presence and absence. The immune cell types examined include naïve B-cells, memory B-cells, plasma cells, CD8 T-cells, CD4 naïve T-cells, CD4 memory resting T-cells, CD4 memory activated T-cells, follicular helper T-cells, regulatory T-cells, gamma-delta T-cells, resting NK cells, activated NK cells, monocytes, M0-M2 macrophages, resting dendritic cells, activated dendritic cells, resting mast cells, activated mast cells, eosinophils, and neutrophils.

### 5.11. Correlation of Microbial Abundance to Cancer and Immune-associates Signatures

The signature enrichment corresponding to microbial abundance was measured using the Geneset Enrichment Analysis (GSEA). Cancer and stem cell-associated signatures were chosen from the C6 set of signatures from the Molecular Signatures Database (MSigDB) [27]. Immune-associated signatures were chosen from the C7 set of signatures. Significantly enriched signatures were identified by a nominal enrichment score > 1 and a nominal *p*-value < 0.05. Canonical (CP) pathways were also included from the C2 set of signatures. The direction of pathway enrichment was filtered to match the direction of clinical variable correlations per microbe.

### 5.12. Correlation of Microbial Abundance to Copy Number Variations and Mutations

The copy number variation (CNV) and mutation data were obtained from annotation files generated by the BROAD Institute GDAC Firehose on March 31, 2018. The surface-level trends of mutation presence were analyzed by calculating the percentage of patients with each mutation, indicated by a binary value per mutation. The GDAC files were compiled into input files for the repeated evaluation of variables conditional entropy and redundancy (REVEALER) algorithm, which identifies sets of specific CNVs and mutations that are most likely implicated in changes to the target expression profile. The target profile was identified as the expression of a single immune-associated gene. The REVEALER algorithm runs in multiple iterations in order to identify the most prominent genomic alterations. For our study, we set the maximum number of iterations to three. The algorithm also allows the use of a seed, or a particular mutation of a CNV gain or loss event that may account for target activity. However, because we did not know the individual genetic alterations that were responsible for each IA gene dysregulation, the seed was set to null. Significant correlations were indicated by *p* < 0.05 and CIC > 0.03.

## Figures and Tables

**Figure 1 cancers-12-02672-f001:**
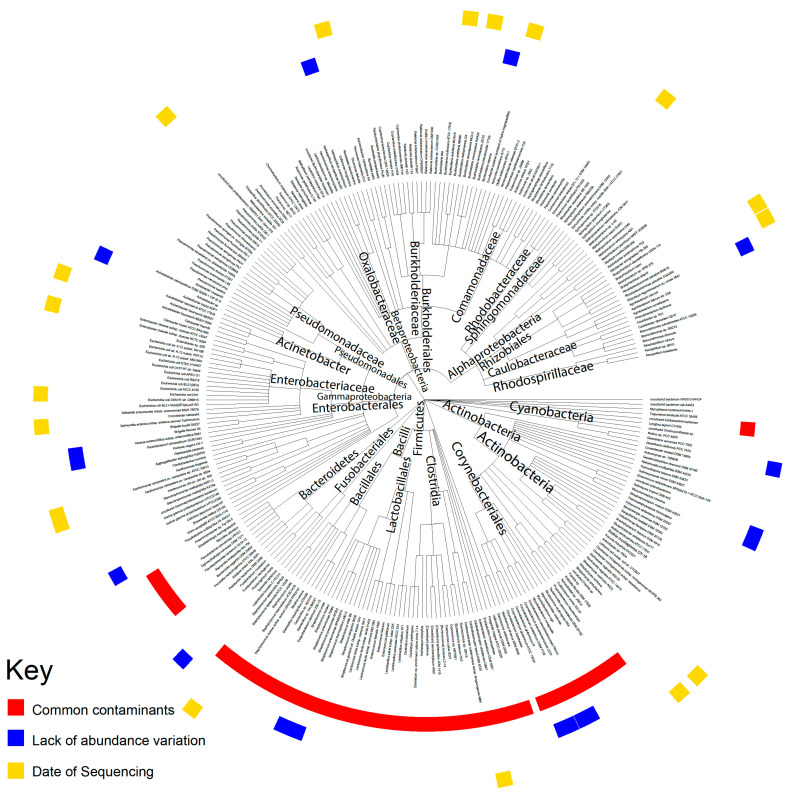
Contamination correction of TCGA data. A summary of the three contamination correction methods used: filtering for common contaminants, microbes with little variation in abundance across patients, and microbes with unusually high abundance corresponding to specific dates.

**Figure 2 cancers-12-02672-f002:**
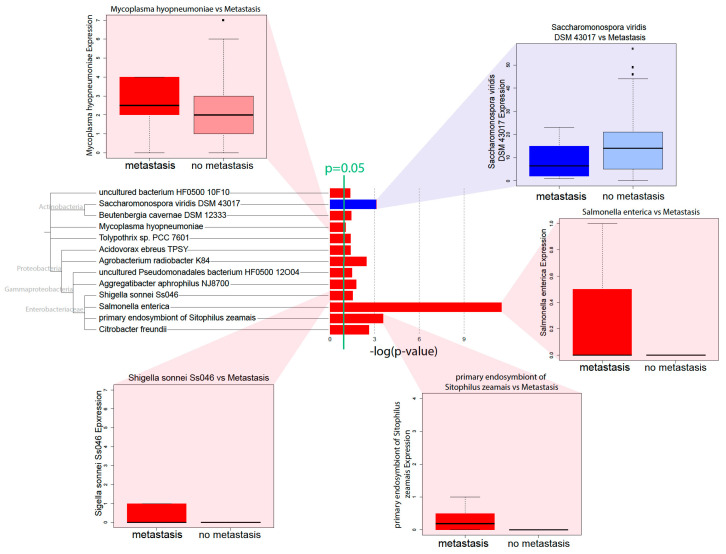
Pathology and survival-associated microbes. Microbial abundance was correlated to metastasis to identify microbes with significant correlations to metastatic tumors (*p* < 0.05). Red bars indicate microbes with a high abundance correlating to metastasis, while blue bars indicate microbes with a low abundance corresponding to metastasis.

**Figure 3 cancers-12-02672-f003:**
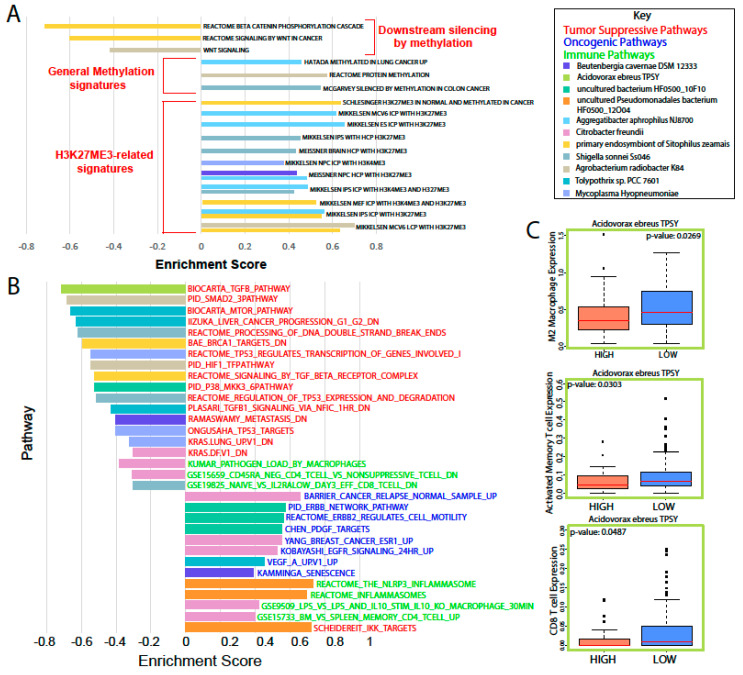
Analysis of cancer and immune-associated gene expression related to microbial abundance. (**A**) The GSEA results indicated a significant number of pathways that were dysregulated to suggest increased methylation that could lead to cancer. (**B**) Immune, tumor suppressive, and oncogenic pathways dysregulated alongside an abundance of pathology or survival-related microbes. (**C**) Immune cell population estimates for patients with a high or low *A. ebreus* abundance.

**Figure 4 cancers-12-02672-f004:**
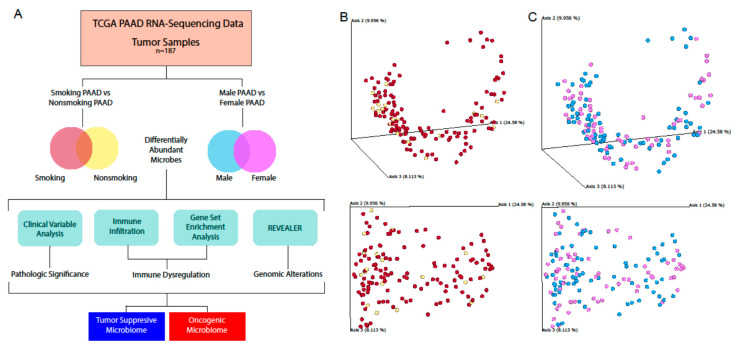
Schematic of etiological analyses. (**A**) A graphical schematic of the analyses used to characterize and investigate the effects of etiology-specific microbiota. The Qiime2 framework was used to calculate the beta diversity and visualize differences between (**B**) smokers and nonsmokers and (**C**) males and females via principle component analysis.

**Figure 5 cancers-12-02672-f005:**
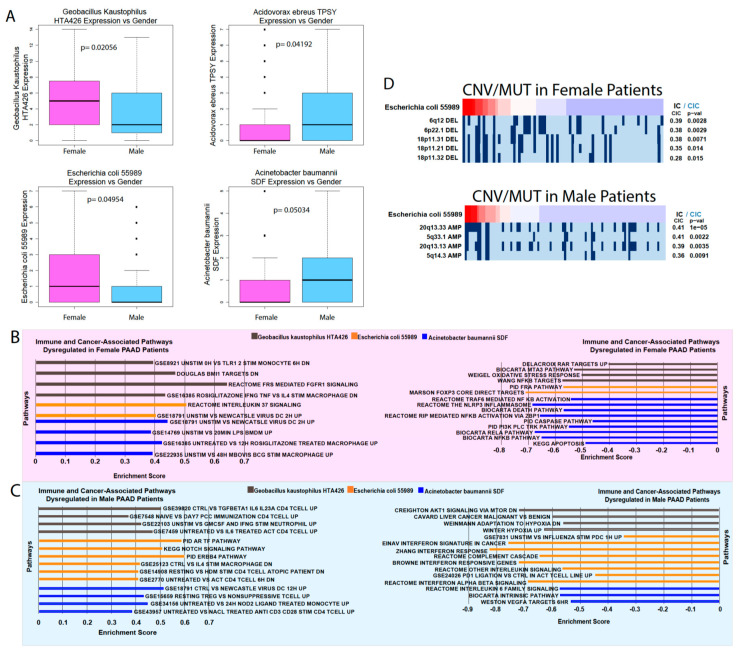
Analysis of the male and female PAAD microbiome. (**A**) Microbes differentially abundant in males and females according to the Kruskal–Wallis test (*p* < 0.05). Immune and cancer-related pathways upregulated and downregulated alongside differentially abundant microbes in (**B**) female and (**C**) male PAAD patients. (**D**) REVEALER analysis results for *E. coli*.

**Figure 6 cancers-12-02672-f006:**
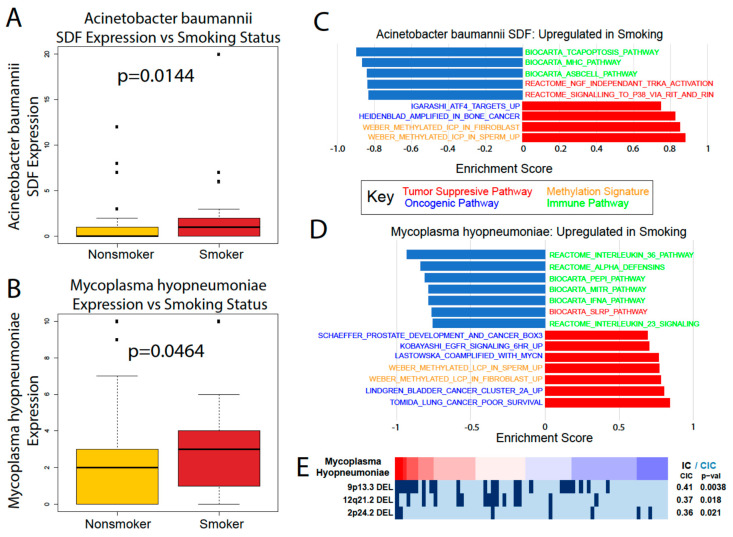
Analysis of the smoking vs. nonsmoking PAAD microbiome. (**A**) *A. baumannii* and (**B**) *M. hyopneumoniae* were both upregulated in smokers versus nonsmokers according to the Kruskal–Wallis test (*p* < 0.05). GSEA analysis comparing the cancer and immune-associated pathways in smokers to the (**C**) *A. baumannii* and (**D**) *M. hyopneumoniae* abundance. (**E**) REVEALER analysis results for *M. hyopneumoniae* in smoking patients.

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
