# Peer review of "The Pancreatic Microbiome is Associated with Carcinogenesis and Worse Prognosis in Males and Smokers"

_cancers, 2020, doi:10.3390/cancers12092672_

Round 1

Reviewer 1 Report

Chakladar and colleagues presented an interesting and original computational study aimed to discover the microbiota signature of pancreatic cancer thus identifying specific bacteria involved in PAAD metastasis and prognosis. The authors used a robust computational approach applying different software in order to minimize confounding results related to contaminants and other variables. Overall, the study is well-conceived and the manuscript well-written, however, there are some aspects that the authors should clarify before publication:

1) In section 2.1 the authors state “Microbes well known to be common contaminants were removed first.”. Please indicate at least the families or genera removed;

2) Figure 1B is not understandable. Consider resubmitting Figure 1B as supplementary materials increasing the size.

3) In Figure 2 it is not clear if the data obtained are statistically significant. Are there statistical differences between metastasis and no metastasis groups for the bacteria 13 bacteria reported in Figure 2?

4) Please check and correct the following sentence “Previous studies have shown the significant impact certain behaviors and characteristics can have on the composition of the microbiome.”;

5) Have the authors tested microbial differences between high-grade and low-grade PAAD patients?

6) In section 2.4 the authors should confirm that the differences in microbiota between male and female is strictly linked to PAAD and not to other confounding factors (e.g. female hormones). Therefore, the authors should filter the results obtained from male and female PAAD patients with those obtained from male and female individuals without cancer;

7) The authors acquired data in august 2018. Have the number of samples, the data or the clinical-pathological features been updated in these two years? Please, clarify and update the data if necessary;

8) In the caption of section 4.7, the authors state “Differential microbial abundance between cancer and normal patients”. However, they only describe the differential abundance analyses performed in cancer tissues stratifying samples according to gender and smoking status. Have the authors analyzed normal individuals? Please clarify, this is a critical issue;

9) The results should be clearly summarized in the conclusion remarks;

10) As previously stated, the lack of normal individuals represents a limitation of the study. Are there microbiome data on normal pancreas? (e.g. data contained in GTEx repository or other databases);

11) Throughout the manuscript there are some errors (e.g. "dyregulated" in Figure Legend 3, "uncultured Pseudomonadales bacterium" the word uncultured should be not in italics; "primary endosymbiont of S. zeamais" primary endosymbiont should be not in italics). Please check carefully all the errors in the text.

Reviewer 2 Report

Pancreatic cancer is one of the only few cancers where distinct associations with local microbiota has been shown. However, the data are still observational and correlative; a cause effect relationship is still unknown. The manuscript “The pancreatic microbiome is associated with carcinogenesis and worse prognosis in males and smokers” is an impressive effort to understand microbiome as a potential pancreatic cancer risk factor. Cancer microbiome studies have always been limited by sample size and inter-individual variations. Multiple studies have analyzed fecal samples which is reflective of the gut microbiota since it is harder to obtain tumor biopsy samples. Using RNA sequencing data from TCGA is a smart way to overcome all these shortcomings since ample data is available for retrospective studies, in addition, samples from different geographical locations across different ethnicities etc can be pooled together to determine solid microbial variations while reducing sequencing expenses. As aptly performed by the authors, microbial signatures can be correlated with gene expression data providing a clear picture of how microbiota changes can alter gene expression profiles. True to what is claimed by the authors, this is first study associating clinic-pathological aspects with tumor microbiota changes in a large group of pancreatic cancer patients. However, in order to be used as a reliable guide for further studies, the authors should clarify the following:

  1. They mention that they obtain microbial signatures from TCGA RNA sequencing data. How exactly was this done? How did they validate the accuracy of their results? Details should be provided in the methodology.
  2. They use 187 patient sample. Was that all that was available or did they use any selectin criteria?
  3. How did they examine “individual sequencing pates” for possible contamination from a database?

Suggestion:

  1. The findings from such in silico studies should always be validated in a smaller sample of patients if possible. Alternatively, microbiome manipulation in animal models for cancer is also a valid methodology.
